# Unexpected CD5^+^ B Cell Lymphocytosis during SARS-CoV-2 Infection: Relevance for the Pathophysiology of Chronic Lymphocytic Leukemia

**DOI:** 10.3390/jcm12030998

**Published:** 2023-01-28

**Authors:** Andrea Nicola Mazzarello, Brisejda Koroveshi, Daniela Guardo, Lorella Lanza, Fabio Ghiotto, Silvia Bruno, Enrico Cappelli

**Affiliations:** 1Department of Experimental Medicine, University of Genoa, Via De Toni 14, 16132 Genova, Italy; 2Laboratory of Clinical Pathology, ASL2 Liguria, S. Paolo Hospital, 17100 Savona, Italy; 3Haematology Unit, IRCCS Istituto Giannina Gaslini, Via Gerolamo Gaslini 5, 16148 Genova, Italy; 4Anatomical Pathology, ASL2 Liguria, Santa Corona Hospital, 17027 Pietra Ligure, Italy; 5IRCCS Ospedale Policlinico San Martino, 16132 Genoa, Italy

**Keywords:** SARS-CoV-2, COVID-19, lymphocytosis, CLL, BCR, TLR, CD40

## Abstract

Recently, cases of fortuitous discovery of Chronic Lymphocytic Leukemia (CLL) during hospitalization for Coronavirus disease (COVID-19) have been reported. These patients did not show a monoclonal B cell expansion before COVID-19 but were diagnosed with CLL upon a sudden lymphocytosis that occurred during hospitalization. The (hyper)lymphocytosis during COVID-19 was also described in patients with overt CLL disease. Contextually, lymphocytosis is an unexpected phenomenon since it is an uncommon feature in the COVID-19 patient population, who rather tend to experience lymphopenia. Thus, lymphocytosis that arises during COVID-19 infection is a thought-provoking behavior, strikingly in contrast with that observed in non-CLL individuals. Herein, we speculate about the possible mechanisms involved with the observed phenomenon. Many of the plausible explanations might have an adverse impact on these CLL patients and further clinical and laboratory investigations might be desirable.

## 1. Introduction

The severe acute respiratory syndrome coronavirus 2 (SARS-CoV-2) infection that causes the coronavirus disease 2019 (COVID-19) has a broad clinical heterogeneity, ranging from asymptomatic to hospitalization, need for mechanical ventilation, and death [1]. During the past two and half years, several studies have proven an association between the individual immune response and the clinical outcome [1,2,3]. Indeed, patients can be grouped based on immunotypes of cellular and molecular responses to the viral infection that are associated with the clinical outcome [1]. Of these immunotypes, thosewhose immune system cannot properly respond (i.e., individuals with immunodepression or immunodeficiency) are at higher risk of hospitalization [4]. Intriguingly, patients with the highest markers of immune response were more often hospitalized [1]. Hence, during SARS-CoV-2 infection, an adequate immunological response is necessary, while divergence from proper activation (i.e., both hyper- or hypo-immune activation) increases the risk of serious illness.

Most COVID-19 patients display overt lymphopenia (Figure 1, top left), which is associated with CD4^+^ and CD8^+^ T cell activation and proliferation, cytokine serum levels increase (e.g., IL-1RA IL-6, IL-8, IL-10, and CXCL10), and T-bet^+^ with decreased CXCR5 B cells [1,3]. Hence, SARS-CoV-2 infection can trigger a complex crosstalk of the immune system.

This complex and heterogeneous immune response to SARS-CoV-2 infection might be further exacerbated in patients with oncological malignancies whose immune system is compromised [5].

Chronic lymphocytic leukemia (CLL), the most commonB cell leukemia in adult Caucasians, is characterized by the expansion of a monoclonal population of CD5^+^ B lymphocytes. As of today, it is not fully clear whether SARS-CoV-2 infection in CLL patients leads, on average, to worse or milder COVID-19 symptomatology compared to age-matched healthy individuals or groups with other comorbidities. Both options are reported and appear linked to (i) the CLL stage, (ii) undergoing CLL treatment, and (iii) the type of treatment [4,6,7,8]. Similarly, the effects of SARS-CoV-2 infection on CLL pathogenesis and clinical evolution are also unclear. Herein, we specifically address the latter issue, based on the available evidence, and try to propose the most likely mechanisms involved.

## 2. Evidencesof Unexpeceted Lymphocytosis of CLL B Cells during COVID-19

We recently described a clinical case that presented to the emergency with SARS-CoV-2 infection symptoms [9]. The patient was admitted with a normal blood cell count. Two days later, a rapid and unexpected increase inthe lymphocyte count led to further investigations that highlighted a population of CD5^+^ monoclonal B cellswith a count above clinical parameters for chronic lymphocytic leukemia (CLL), leading to a new CLL diagnosis [9].

After searching the scientific literature, we realized that anecdotal cases of fortuitous discovery of CLL during hospitalization for COVID-19 had also been reported by others [9,10,11,12]. These patientsdid not show a monoclonal B cell expansion before COVID-19 but were diagnosed with CLL upon a sudden lymphocytosis that occurred during hospitalization. The lymphocytosis associated with the emergence of a CD5^+^ monoclonal B cell expansion is referable to CLL (Figure 1, top middle) [9,10,11,12]. In all these cases, the leukemic clone persisted after recovery from COVID-19 and partial resolution of lymphocytosis. Thus, CLL diagnosis was confirmed, except for one case that turnedinto Monoclonal B lymphocytosis (MBL) [9].

Cases of infection triggering the earlier unmasking of CLL were also reported in a Swedish study on the outcome of COVID-19 in CLL patients, where 8% (5/60) of enrolled cases were diagnosed with CLL during hospitalization. Yet, no information on lymphocytosis phenomena is available since the CLL blood count over time was not reported [6]. Thus, the true number of new CLL cases associated with this transient lymphocytosis is currently unknown. However, a significant number of anecdotal cases display this association [9,10,11,12].

Interestingly, the (hyper)lymphocytosis phenomenon was also described in patients with overt CLL disease (Figure 1, top right) [13,14,15,16]. Likewise, the percentage of CLL patients whoundergo (hyper)lymphocytosis during SARS-CoV-2 infectionis unknown. Nevertheless, there is a consensus among hematologists that CLL patients “often”display this phenomenon.

Contextually, lymphocytosis is an unexpected phenomenon since it is an uncommon feature in the COVID-19 patient population, whorather tend to experience lymphopenia (Figure 1, top left) [1,3]. Thus, lymphocytosis thatarises during COVID-19 infection is a thought-provoking behavior, strikingly in contrast with that observed in non-CLL individuals (Figure 1, “Observation”) [1,3].

What are the causes that trigger the (hyper)lymphocytosis of the leukemic cells? Additionally, how does (hyper)lymphocytosis, and more in general, SARS-CoV-2 infection, affect CLL disease in the longterm?

## 3. Plausible Explanations of the Observed CLL B Cellslymhocitosis during COVID-19 

Generally, infections and associated inflammatory responses are believed to have a role in the pathogenesis of some CLL cases [17,18]. Yet, CLL cells modulate various components of the immune system, such as T helper (Th) cells and Myeloid-Derived Suppressor Cells (MDSCs), suppressing the anti-tumor response against themselves and creating a trophic microenvironment for their own support [19,20]. It is possible that an immune response against SARS-CoV-2 might lead to various activations and modulations of the immune system that might be either advantageous or deleterious for the leukemic B cells.

Time is needed for comprehensive and data-supported answers. However, we can speculate about the mechanism(s) beyond this phenomenon and the possible acute and chronic effects on the CLL.

The relatively quick and high rise inthe CLL cells’ blood count likely excludes (hyper)lymphocytosis dueto the induced proliferation of leukemic cells. More likely, through various mechanisms evaluated below, the infection might induce outwards trafficking of CLL cells from the secondary lymphoid organs.

Normally, most peripheral CLL cells are quiescent and/or anergic, while those residing in the secondary lymphoid tissues (e.g., lymph nodes, spleen, bone marrow) compose populations of active cells responsive to stimuli and with a significant percentage of proliferating cells [21,22,23]. The models of in vivo kinetics of the CLL generally propose that in the secondary lymphoid organs, CLL B cells undergoing activation and/or proliferation downregulate adhesion and trafficking molecules (e.g., CXCR4) and migrate outwards [21]. Once in the bloodstream, they acquire less active, more quiescent/anergic phenotypes while concomitantly re-upregulating trafficking molecules [21]. A fraction of these CLL cells will traffic inwards toward the secondary lymphoid organs, perpetuating the cycle [21,22]. When the rates of survival/proliferation overcome those of apoptosis, CLL can progress.Hence, it is possible that during the acute phase of SARS-CoV-2 infection, part of the CLL cells residing in the secondary lymphoid tissues might undergo activation, either directly (e.g., viral antigens engaging the BCRs or TLRs of the CLL cells) or indirectly (e.g., activation of T cells and the proinflammatory response that support CLL B cells’ activation/proliferation). All these mechanisms are known to downregulate trafficking receptors such as CXCR4 on the CLL cells, leading to outward trafficking and (hyper-)lymphocytosis [24,25,26,27].

Concerning the possible direct engagement through the BCR, CLL is known to often display polyreactive antibodies against self- and (non)self-antigenswith a key role both during CLL pathogenesis and clonal evolution [17,18,28,29,30]. Moreover, cases of CLLs expressing antibodies against bacteria or fungi identified in previous infections have beenreported [17,18]. Possibly, a fraction of CLL patients might be able to express BCRs to directly react with SARS-CoV-2 antigens, therefore undergoing activation/proliferation and being released from secondary lymphoid tissues (Figure 1, bottom left). However, this is unlikely to be the most common cause of lymphocytosis since it is observable in CLL cases expressing different antigen-binding sites.

Direct activation of CLL cells might rather be inducedthrough the TLR7 pathway asCLL cells are activated through the TLRs (e.g., TLR7), and RNA from SARS-CoV-2 has beenfound in the lymph nodes and spleen of infected individuals (Figure 1, bottom left) [2,31].

CD4^+^ T cells are also often activated by SARS-CoV-2 infection and may show increased expression of CD40L among several activation markers [32]. CLL B cells require support from autologous T cells, andinteraction between CD40L and CD40 might induce activation and proliferation [33,34]. Additionally, IL-4, an important co-factor for the CD40-mediated CLL activation, is upregulated during SARS-CoV-2 infection in a significant fraction of cases [32]. Hence, SARS-CoV-2 infection might indirectly activate CLL cells and induce the subsequent downregulation of trafficking receptorsand lymphocytosis by activating the CD4^+^ T cells’response against the virus(Figure 1, bottom left).

Conversely, SARS-CoV-2-mediated activation of T cells and, more in general, of the entire immune system, might be the mechanism by which CLL cells lose the trophic nicheswithin the secondary lymphoid organs, thus causing a passivelymphocytosis of quiescent cells, which represents a favorable process for the CLL patient (Figure 1, bottom right).

## 4. Conclusions

Altogether, the response to COVID-19 among CLL patients is highly heterogeneous. The kind of response to the infection and the features of the specific CLL clone likely co-play inthe ultimate CLL cell’s fate (Figure 1, “Hypothesis”). Notably, the BCR, TLR9, and CD40 signaling pathways may share common key signaling checkpoints [35,36]. Overall, concomitant crosstalk and influence from more than one pathway arelikely. 

Especially, as proliferation is linked with an increased likelihood of errors during DNA synthesis, COVID-19 increases the chance of novel genetic mutations upon activation of CD5^+^ neoplastic B cells. The former possibility leads to the question of whether COVID-19 might also exacerbate underlying leukemia that is still below the clinical parameters for a positive diagnosis, as some recent reports havepointed to [9,10,11,12]. As of now, it is too early to clinically evaluate a change in the CLL outcome upon COVID-19 diagnosis. Similarly, lab tests have not yet fully investigated the phenotype of CLL B cells during induced (hyper)lymphocytosis. Thus, future analyses are required to examine the activation/exhaustion of leukemic cells, helping to clarify which mechanism(s) are involved and considering possible changes in CLL therapy during and after SARS-CoV-2 infection.

## Figures and Tables

**Figure 1 jcm-12-00998-f001:**
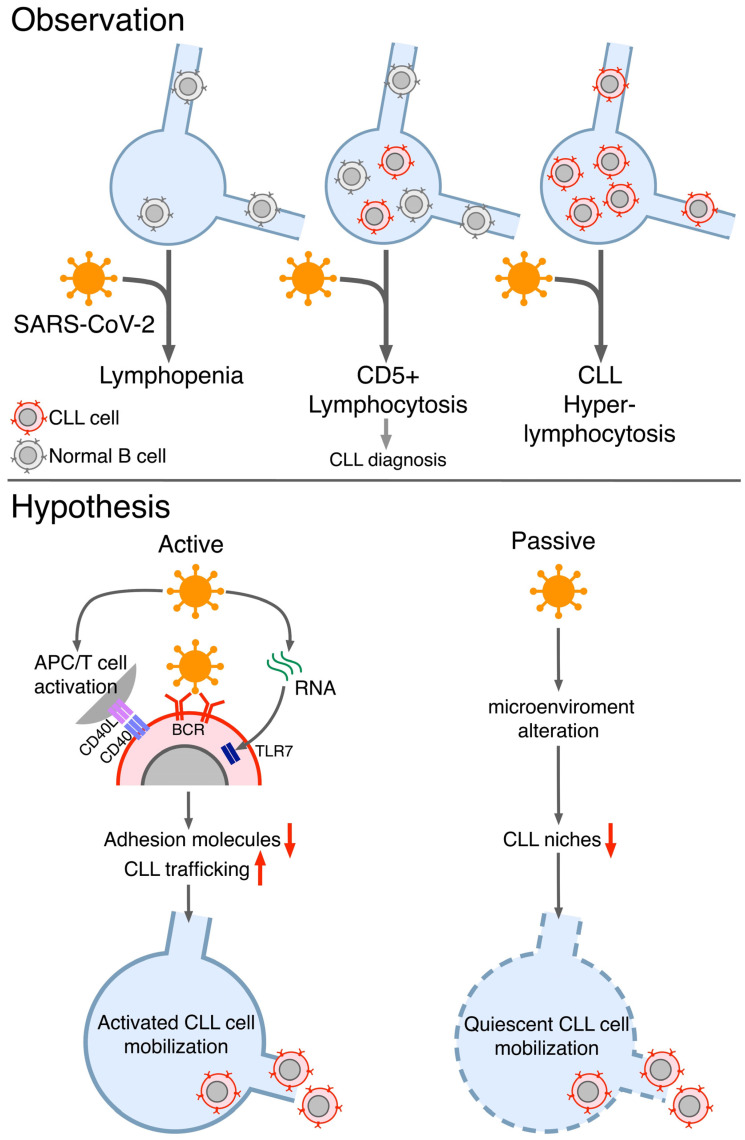
Schematic representations of blood-count discrepancies between healthy subjects and CD5^+^ B cell patients upon SARS-CoV-2 infection and the possible mechanism(s) induced by SARS-CoV-2 that may drive (hyper)lymphocytosis of either active or quiescent CLL B cells. During SARS-CoV-2 infection, healthy individuals generally displayed lymphopenia (**top left**). In contrast, some apparently healthy subjects underwent a sudden CD5^+^ B cells lymphocytosis whose CLL origin was then unveiled (**top middle**). Likewise, patients with overt CLL often responded to SARS-CoV-2 with (hyper)lymphocytosis (**top right**). These observed CLL (hyper)lymphocytosis cases in the two latter populations might be due to SARS-CoV-2 engaging several pathways associated with CLL activation response and leading to the mobilization of activated leukemic cells (**bottom left**). On the other side, the broad and potent immune response upon infection was demonstrated to change the architecture and biological properties of the CLL niches in the secondary lymphoid organs, leading to the release of quiescent cells into the bloodstream (**bottom right**).

## Data Availability

Not applicable.

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
