# Peer review of "Unexpected CD5+ B Cell Lymphocytosis during SARS-CoV-2 Infection: Relevance for the Pathophysiology of Chronic Lymphocytic Leukemia"

_jcm, 2023, doi:10.3390/jcm12030998_

Round 1

Reviewer 1 Report

Nice and exhaustive paper developing an"hypothesis", that potentially links B cell activation, B cell clonal expansion and B-CLL pathogenesis in COVID19 infection setting.

Only one point to address: has this sudden B cell expansion been observed in other human viral infections? More specifically, it has been observed in other respiratory infections sustiained by a virus? Probably this point is worth an enlarged discussion.

Author Response

Reviewer 1, Comment: “Nice and exhaustive paper developing an "hypothesis", that potentially links B cell activation, B cell clonal expansion and B-CLL pathogenesis in COVID19 infection setting. Only one point to address: has this sudden B cell expansion been observed in other human viral infections? More specifically, it has been observed in other respiratory infections sustained by a virus? Probably this point is worth an enlarged discussion.”

We thank the reviewer for this important question. It is possible that other infections might lead to similar clinical behavior(s). However, at the best of our knowledges, there are no published data about (hyper)lymphocytosis of CLL cells during respiratory or non-respiratory viral infections, neither as large studies nor as anecdotal case reports. We believe that the health care attention developed for the SARS-CoV-2 pandemic, has allowed to more easily identify an ongoing infection that induces CLL (hyper)lymphocytosis.

Reviewer 2 Report

In this short paper, Mazzarello and co-workers presented a series of hypotheses to explain an unusual lymphocytosis in patients with SARS-CoV-2 infection and a concomitant diagnosis of CLL.

The paper is mainly speculative although based on published biological observations.

To facilitate the interest of potential readers, a figure summarizing the different biological aspects could help.

Author Response

Reviewer 2 Comment: “In this short paper, Mazzarello and co-workers presented a series of hypotheses to explain an unusual lymphocytosis in patients with SARS-CoV-2 infection and a concomitant diagnosis of CLL. The paper is mainly speculative although based on published biological observations. To facilitate the interest of potential readers, a figure summarizing the different biological aspects could help.”

Authors reply. We agree with the reviewer’s comment and are thankful for the suggestion. To this end, we have included a schematic figure representing the observations and the hypotheses formulated to explain the hematological differences of the response to SARS-CoV-2 observed in healthy subjects and undiagnosed or overt CLL patients. A brief figure legend was included to facilitate reading of the Figure.